# Effects of Constant versus Fluctuating Temperatures on Fitness Indicators of the Aphid *Dysaphis plantaginea* and the Parasitoid *Aphidius matricariae*

**DOI:** 10.3390/insects12100855

**Published:** 2021-09-23

**Authors:** Kévin Tougeron, Louise Ferrais, Marie-Eve Renard, Thierry Hance

**Affiliations:** 1Earth and Life Institute, Ecology and Biodiversity, Université Catholique de Louvain, 1348 Louvain-la-Neuve, Belgium; louise.ferrais@uclouvain.be (L.F.); marie-eve.renard@uclouvain.be (M.-E.R.); thierry.hance@uclouvain.be (T.H.); 2UMR CNRS 7058 EDYSAN, Écologie et Dynamique des Systèmes Anthropisés, Université de Picardie Jules Verne, 80000 Amiens, France

**Keywords:** rosy apple aphid, biological control, integrated pest management, microclimate, variation

## Abstract

**Simple Summary:**

Like all organisms, insects encounter temperatures that fluctuate on different time scales: within a day, between days, or throughout the seasons. However, most studies on the impact of temperature on insect physiology, behavior, morphology, or ecology have focused on constant temperatures tested in the laboratory. In our study, we wanted to know if fluctuating temperatures during the day (7–17 °C, average 12 °C) can affect insects differently compared to a constant temperature of 12 °C. We used, as a model, the apple aphid *Dysaphis plantaginea*, a major threat to apple orchards worldwide, and its parasitoid *Aphidius matricariae*, which is used in biological control. We found that many traits—but not all—were affected. In particular, the fluctuating thermal regime reduced the development time of aphids and parasitoids, improved the rate of parasitism, and tended (albeit slightly) to increase the longevity of both species. In contrast, we did not find strong effects on morphological traits. Our results can be used to better predict how these agronomically important insects behave in orchards, how ecologically-relevant fluctuating temperatures affect host–parasitoid relationships, and ultimately what the implications are in the context of climate change and biological control.

**Abstract:**

Testing fluctuating rather than constant temperatures is likely to produce more realistic datasets, as they are ecologically more similar to what arthropods experience in nature. In this study, we evaluated the impact of three constant thermal regimes (7, 12, and 17 °C) and one fluctuating thermal regime (7–17 °C with a mean of 12 °C) on fitness indicators in the rosy apple aphid *Dysaphis plantaginea*, a major pest of apple orchards, and the parasitoid *Aphidius matricariae*, one of its natural enemies used in mass release biological control strategies. For some—but not all—traits, the fluctuating 7–17 °C regime was beneficial to insects compared to the constant 12 °C regime. Both aphid and parasitoid development times were shortened under the fluctuating regime, and there was a clear trend towards an increased longevity under the fluctuating regime. The fecundity, mass, and size were affected by the mean temperature, but only the mass of aphids was higher at 7–17 °C than at a constant 12 °C. Parasitism rates, but not emergence rates, were higher under the fluctuating regime than under the constant 12 °C regime. Results are discussed within the framework of insect thermal ecology and Jensen’s inequality. We conclude that incorporating thermal fluctuations in ecological studies could allow for the more accurate consideration of how temperature affects host–parasitoid interactions and insect responses to temperature change over time.

## 1. Introduction

In apple orchards, a major insect pest is the rosy apple aphid *Dysaphis plantaginea* (Hemiptera: Aphidinae), which causes leaf-rolling, fruit deformation, and consequent yield losses [1]. The population growth rate of this species is quite low [2], but the main threat comes from the phenology of the aphid. In early spring, fundatrix females hatch from overwintering eggs during the bud break, which is followed by several generations of wingless parthenogenetic viviparous females in the spring and early summer, responsible for most of the damages on apple trees [2]. Targeting fundatrix aphids and the first following generation in early spring is crucial for efficient control strategies, because they are the starting point of an exponential and massive parthenogenetic reproduction that will be damaging to trees. Apple production relies on a heavy use of pesticides to control different kinds of pests, including the rosy apple aphid [3,4]. In the past decades, concerns about the risks associated with pesticides, the development of chemical-resistant pest strains, and public growing demand for organic food production have boosted the search for economically and ecologically sustainable solutions for pest control in orchards, stimulating interest towards natural enemies [5].

A set of predator arthropods is active in apple orchards: aphid midges, predatory mites, hoverflies, earwigs, ladybugs, and spiders are among the most abundant predatory arthropods that help to reduce populations of the rosy aphid [5,6]. Naturally occurring parasitic wasps in orchards, mostly Braconidae, are also reducing the *D. plantaginea* population growth [7,8,9]. However, most of the natural enemies occur too late in the aphid’s lifecycle to exert strong regulating effects on large colonies formed by fundatrix aphids [6,7]. For this reason, in organic production, predators and parasitoids often struggle to reduce the aphid abundance below the economic threshold [1,10], especially in years of high aphid density [11]. Therefore, augmentative releases of beneficial insects in early spring have been proposed to complement the impact of the naturally occurring aphid control agents [12,13].

As part of augmentative biological control strategies, a mix of two solitary parasitoid species, *Aphidius matricariae* and *Ephedrus cerasicola* (Hymenoptera: Braconidae), was proposed to control early infestations of *D. plantaginea* in apple orchards [14,15,16]. Pilot field studies showed a poor control of the rosy aphid, probably because of the high heterogeneity of the parasitoid establishment after introduction, which may be due to the relatively low temperatures encountered in orchards in early spring [17]. Preliminary laboratory results on parasitoid behavior demonstrated that *A. matricariae* was not able to fly below 10 °C constant temperatures, and that walking capacities and parasitism rates were strongly reduced below 15 °C [18]. The reduced activity of the released parasitoids may jeopardize aphid control. To develop accurate biological pest control strategies in orchards, it is therefore crucial to better understand how both the aphid pest and its natural enemies respond to temperature conditions encountered at the early stage of the aphid population outbreak.

Poikilothermic organisms, such as insects, are, by definition, very sensitive to temperature changes in space and time, and thermal performance curves (TPC) allow for the description of the effects of temperature on performance [19]. Historically, the assessment of behavioral and physiological responses of *A. matricariae* [18,20,21,22] and *D. plantaginea* [2,23] to various temperatures has been mostly performed at constant temperatures. Globally, and according to the studies mentioned above, the response of these insects follows a typical thermal performance curve, with a relatively low performance (e.g., fecundity) at a low temperature, a performance optimum between 17 and 23 °C, and a sharp decrease in the performance at higher temperatures. The lower development threshold is 4.5 °C in *D. plantaginea* and around 5 °C in *A. matricariae* [20,23]. The longevity usually increases at non-stressful low temperatures due to a reduction in metabolic rates [19]. In addition, as a general temperature–size rule in insects, bigger adults are formed if they have developed under lower temperatures because of reduced growth rates leading to longer development times [24,25]. Host–parasitoid interactions, such as exploitation behaviors, parasitism rates, network composition, and relative abundances, are also strongly influenced by temperature [25,26,27].

However, insects respond differently to constant versus fluctuating temperature regimes, so conclusions drawn from data sets generated from studies at constant temperatures may not be fully reliable in an applied biological control perspective [28,29]. As predicted by the Jensen’s inequality (i.e., the asymmetric shape of TPCs describing the nonlinear relationship between temperature and life history and ecological processes), fluctuating environments lead to consequences for ectothermic insects that diverge from those predicted at constant temperatures [28]. Thermal fluctuations can improve the insect performance relative to a constant temperature when conditions remain within permissive temperature ranges, or, for example, by allowing for a recovery from damages from thermal extremes. However, it can also have negative or neutral impacts depending on the focus trait or process, the amplitude of the fluctuating temperature regime, or the mean temperature value relative to the critical thermal limits of the TPC (reviewed by Colinet et al. [29]).

Numerous studies demonstrate how insect survival, morphology, fecundity, and the development period are affected by temperature fluctuations, either positively or negatively, eventually affecting species interactions and the provision of ecosystem services [30,31,32,33]. For example, Bayu et al. [34] showed that the intrinsic rate of population increase in the spider mite *Tetranychus urticae* was higher under a fluctuating 15 °C (±10 °C) thermal regime, compared to a constant 15 °C, but that the temperature fluctuation was not always favorable for other traits or other regimes. In the parasitoid *Aphidius colemani*, Colinet et al. [35] demonstrated that applying periodic transfers from 4 to 20 °C for 2 h significantly improved the survival of immature insects, probably due to their physiological recovery during this time.

To our knowledge, a comparison between the effects of constant versus fluctuating temperatures on the *D. plantaginea*—*A. matricariae* association has not been conducted. In this study, we analyzed the effects of three constant thermal regimes and one fluctuating thermal regime on the fecundity, survival, and morphological traits and the interaction of both parthenogenetic viviparous females *D. plantaginea* (damaging stages) and *A. matricariae*. Tested mean temperatures represent thermal conditions that are likely to be experienced in apple orchards in Northwest Europe at the early stage of aphid infestation in early spring, and the fluctuating regime allows for the testing of more naturally accurate temperatures. One of our goals was to see whether fluctuating regimes would increase or decrease the parasitoid and aphid fitness and performance, compared to what is predicted from constant regimes. In orchards, fluctuating spring temperatures could allow parasitoids to take advantage of occasional peaks of heat and perhaps become as effective in controlling aphids and as performant in terms of traits as when under constant conditions. We expected aphids and parasitoids to have a longer development time, to survive longer, and to be bigger at cold temperatures, and parasitism rates to be higher at a higher temperature. However, for a given mean temperature, we expected the value of these traits to be more advantageous for the insects under fluctuating regimes than under constant regimes.

## 2. Material and Methods

### 2.1. Biological Material

Apple trees (*Malus domestica* v. Jonagold) were obtained from the CRA-W (*Centre wallon de Recherches Agronomiques*, Gembloux, Belgium). They were cultivated in potting soil and one fungicide application was made during the early growing stages of the plant. Aphids *D. plantaginea* were collected in 2018 in apple orchards in Wallonia (Belgium) and were maintained on apple trees. Parasitoids *A. matricariae* were purchased from Biobest (Westerlo, Belgium) and were maintained on a laboratory strain of the aphid *Myzus persicae* on sweet pepper plants, which is more convenient to maintain in cultures than *D. plantaginea*. Previous work on this system showed that parasitoids had no problem switching host species [18]. Plants and insects were all maintained at 18 ± 1 °C, 16:08 h light:dark photoregime, and 60 ± 15% relative humidity before the experiments.

### 2.2. Experimental Design

#### 2.2.1. Thermal Regimes

Four thermal treatments were used: constant 7 °C, constant 12 °C, constant 17 °C, and a fluctuating regime of 7–17 °C (12 h), 17–7 °C (12 h), under a 12:12 h photoregime, corresponding to a mean temperature of 12 °C over the day and a rate of temperature change of 1.66 °C per hour. For the fluctuating regime, the 12 h increasing temperature phase started at midnight and the 12 h decreasing temperature phase started at midday, resulting in a triangle-like gradual decrease and increase regime. These conditions were chosen because they correspond to the minimum, mean, and maximum temperatures encountered in Wallonia (Louvain-la-Neuve, Belgium) on 25 March 2019 (Royal Meteorological Institute of Belgium). Around this date, apple trees undergo leaf budburst and are very sensitive to aphid infestation. The four treatments were completed in the same climate chamber (MLR-352H, PHC Europe, Etten-Leur, The Netherlands) over a period of four months at 70% ± 10% relative humidity. Aphids and parasitoids were habituated 72 h to the appropriate thermal regime before the start of the experiments by putting rearing cages directly in the climate chamber.

#### 2.2.2. Aphids

We tested the effect of the different thermal regimes on aphid survival and on their fecundity. We focused on parthenogenetic viviparous females, which are the morphs that cause most of the damages in spring in apple orchards. Three young adult female aphids were taken from the colony and individually put on apple tree leaves placed in glass petri dishes (Ø 10 cm) on a 1.5% agar substrate for larviposition, at 20 °C. Soon after emergence, larvae were gently removed from the parental petri dish, individually placed on a new leaf, and immediately put under one of the tested thermal regimes. This was carried out for a total of four larvae per aphid mother, resulting in 12 tested aphids coming from three different parental genotypes, and for each thermal regime. Apple tree leaves were replaced every ten days to avoid any effect of dehydrated leaves or of fungal infection of the agar substrate. We measured the duration of the pre-reproductive period (recorded as “NA” for aphids that died during immature stages), the total longevity, the rate of immature mortality (i.e., death of the aphid before first larviposition), and the total number of larvae produced per female [2]. For each temperature regime, four additional aphids from five parental lines (i.e., a total of 20 per treatment) were kept separately and placed in a freezer at −20 °C the day of their molt to adult, for later trait measurements.

#### 2.2.3. Parasitoids

For each temperature treatment, around 30 third larval instar aphids were placed on an apple leaf on agar substrate in a glass petri dish, and aphids were allowed to set up on the leaf for two hours before parasitoid introduction. One mated female parasitoid *A. matricariae*, aged <48 h, was gently introduced into the petri dish and allowed to parasitize aphids. Parasitism occurred in the climate chamber under one of the four treatments. A small piece of cotton soaked with 50% diluted honey was placed in the petri dish alongside the apple leaf to feed the parasitoid. After 24 h, the parasitoid was removed from the petri dish and aphids were monitored daily for mummy formation. Once a mummy was formed, it was isolated in a microperforated PCR tube (1.5 mL) to allow air flow, and kept to monitor parasitoid emergence. Therefore, we obtained the parasitism rate (i.e., number of mummies/total number of aphids) and the emergence rate (i.e., number of parasitoids emerging from the mummies/total number of mummies). Among the parasitoids from the next generation, one female and one male (when possible) per petri dish replicate (i.e., per brood) were randomly picked the day of their emergence. The development time (±24 h) from oviposition to adult emergence was noted for these two parasitoids per brood. They were then kept individually in PCR tubes, with access to a drop of 50% diluted honey regularly renewed until they died, to estimate the longevity. The day they died, the parasitoids were stored at −20 °C for future trait measurements. This entire protocol was repeated ten times (i.e., with ten parasitoid mothers) for each temperature regime.

#### 2.2.4. Fitness-Related Traits

We analyzed morphological trait indicators of fitness on a total of 20 adult aphids (four per aphid’s parental line), and 20 adult parasitoids (two per parasitoid’s brood, trying to balance the sex-ratio), for each treatment, except at 7 °C, where only 13 aphids survived to adulthood. The length of the hind right leg tibia, a proxy for adult size, was measured by image analysis method. Digital pictures of tibia were captured with a camera (Sony N50) mounted on a stereomicroscope and pictures were analyzed using the numeric image analysis software ImageJ (Wayne Rasband, Kensington, MD, USA). The insects were also weighted with an electronic precision balance (Mettler-Me22; sensitivity: 1 μg) to obtain dry mass after being dried at 60 °C for 48 h.

### 2.3. Statistical Analyses

Total offspring number produced by aphids were compared among temperature treatments using a generalized linear model (GLM) with a negative binomial family and a log link function. The pre-reproductive period of aphids, the time to adulthood of parasitoids, and the longevity of both species were compared among temperature treatments using Cox models (i.e., survival analysis). Mass and size of aphids and parasitoids were compared among temperature treatments using linear models (normality of residuals and homogeneity or variance were checked).

For parasitoids, the sex of the individual was used as an additional explanatory factor in the different models on trait analysis. Preliminary analyses showed that there was no interaction effect between the temperature regime and the sex of the parasitoid, i.e., that male and female parasitoids showed the same response to the temperature treatment for any parameter. Therefore, and to simplify the message, comparisons of the various measured parameters among temperature regimes were carried out using pooled data of the two sexes, and we provided a comparison between sexes for each temperature regime.

Tukey’s post-hoc tests were performed to reveal pairwise differences between treatments. All statistical analyses were conducted in R version 4.0.3 [36] with the help of the *car*, *survival*, *performance*, and *emmeans* packages [37,38,39,40].

## 3. Results

The pre-reproductive period of aphids was different according to the temperature treatment (Cox model, χ^2^ = 70.79, df = 3, *p* < 0.001). The lower the temperature, the longer it took for aphids to reach their reproductive period. In the fluctuating regime, aphids reached the reproductive period 1.3 times faster than at a constant 12 °C.

The aphid longevity was dependent on the rearing temperature (Cox model, χ^2^ = 15.71, df = 3, *p* < 0.01). It was the lowest at 17 °C and the highest at 7 °C. There were no significant differences between the fluctuating regime and other temperatures.

The total offspring number produced per aphid female was affected by the temperature treatment (GLM, LR = 99, df = 3, *p* < 0.001). The highest number of offspring was produced at 17 °C, whereas the lowest was produced at 7 °C. The fluctuating regime did not differ from the constant 12 °C or 17 °C (Table 1).

The mass (LM, F = 8.16, df = 3, *p* < 0.001) and size (F = 9.74, df = 3, *p* < 0.001) of aphids were significantly influenced by the temperature treatment. Aphids were heavier under the constant 7 °C and the fluctuating 7–17 °C thermal regimes than under the constant 12 °C and 17 °C regimes. They were also smaller at the constant 17 °C regime than under any other temperature regime.

The mass of parasitoids was only marginally affected by the temperature regime (F = 2.48, df = 3, *p* = 0.06) and by the sex of the parasitoid (F = 3.42, df = 1, *p* = 0.06). There were differences in mass only between the constant 17 °C (light parasitoids) and constant 7 °C (heavy parasitoids). Male parasitoids were lighter than females, with a marginally non-significant effect. The size of parasitoid tibias, however, was significantly influenced by the temperature treatment (F = 6.17, df = 3, *p* < 0.001) and by the sex of the parasitoid (F = 9.48, df = 1, *p* < 0.01). The parasitoid tibia size was higher at 7 °C than at 17 °C, but did not differ with other temperature regimes. Male parasitoids were, on average, 1.2 times smaller than females, all temperature regimes considered (Table 2).

Both parasitism rates (GLM, LR = 221.1, df = 3, *p* < 0.01) and emergence rates (LR = 47.9, df = 3, *p* < 0.01) differed according to the temperature regime. Both parameters were the lowest at 7 °C. Parasitism rates were similar between the 7–17 °C fluctuating treatment and the constant 17 °C treatment, and emergence rates were similar between the fluctuating treatment and both the constant 12 °C and 17 °C treatments (Figure 1).

The development time of parasitoids was affected by the temperature treatment (Cox model, χ^2^ = 167.3, df = 3, *p* < 0.001) but not by the sex of the parasitoid (χ^2^ = 2.50, df = 1, *p* = 0.12). For both sexes pooled, the lower the temperature, the longer the development time. In particular, the development time under the fluctuating temperature treatment was significantly shorter than under the constant 12 °C treatment. Means ± SE were as follows: 60.95 ± 1.06, 42.15 ± 0.67, 21.10 ± 0.46, and 38.90 ± 0.66 days, for the 7, 12, 17, and 7–10 °C treatments, respectively (Figure 2a).

The survival probability (longevity) was affected by the temperature treatment (Cox model, χ^2^ = 12.48, df = 3, *p* < 0.01) but not by the sex of the parasitoid (χ^2^ = 0.04, df = 1, *p* = 0.83). For both sexes pooled, the survival probability was the lowest at 17 °C and the highest at 7 °C. A trend showing a longevity increase at the fluctuating 7–17 °C regime compared to the constant 12 °C could be observed. Nevertheless, there was no significant statistical difference in the survival probability between the fluctuating 7–17 °C regime and constant 12 and 17 °C regimes. Mean ± SE longevities were as follows: 43.2 ± 4.1, 35.2 ± 3.7, 29.9 ± 2.2, and 43.1 ± 3.4 days, for the 7, 12, 17, and 7–10 °C treatments, respectively (Figure 2b).

## 4. Discussion

By comparing the effects of three constant thermal regimes and one fluctuating thermal regime on the aphid *D. plantaginea* and the parasitoid *A. matricariae*, our aim was to determine how insect fitness-related traits could respond to ecologically more accurate and relevant temperature variations, as compared to commonly used constant laboratory conditions. In particular, it is interesting to focus on the results comparing the constant 12 °C treatment with the fluctuating 7–17 °C regime (i.e., 12 °C on average over the day). In any case, the fluctuating regime was never a disadvantage for the insects, and we found that, for some—but not all—traits, the fluctuating regime was even advantageous for the insects, as compared to the constant 12 °C regime. Aphids and parasitoids under the 7–17 °C regime could take advantage of the warmest hours of the day to recover from the coldest periods, and improve their fitness.

Globally, our results are consistent with the expected effects of temperature changes on insect life-history traits, regardless of the effects of constant vs. fluctuating regimes. We found, for both the parasitoids and the aphids, a relatively low performance at the lowest tested temperature (7 °C), a higher performance at 17 °C, which can be interpreted as the optimal temperature (among the temperatures tested in our study), and an intermediate performance at the intermediate mean temperature (12 °C) [22,41]. For example, bigger adults were formed under lower temperatures, which could be in part because of longer development times [24]. The behavior of parasitoids was also directly and negatively affected by a decrease in temperature, as shown from the sharp decrease in parasitism rates at 7 °C. It is very likely that 7 °C does not allow for the optimal exploitation of the hosts by parasitoids, which can lack movement coordination and have reduced decision-making capacities at this temperature [18,42]. Although we conducted experiments within a non-lethal temperature range for both species, the lowest tested temperature was close to the developmental threshold of both species, which has been estimated to be at around 4–5 °C [22,23], suggesting that some physiological or behavioral processes could have been impaired if their lower threshold is close to these temperature values. This general pattern has the exception of longevity, which was longer at a low temperature, most likely due to a reduction in insect metabolic rates [19].

Aphids reached the reproductive period and parasitoids developed to the adult stage faster under the fluctuating regime than under the constant 12 °C regime. Fluctuating temperatures that remain within the permissive thermal range of an insect, as is the case in our study, can result in higher developmental rates (i.e., shorter development times) [29,43], although longer development times have already been observed [29]. For example, in *Manduca sexta* (Lepidoptera: Sphingidae), a fluctuating ± 15 °C diurnal regime at a mean temperature of 25 °C shortens mean development time by almost two days (≈25 to 23 days) compared to a constant regime [44]. Due to the Jensen’s inequality (i.e., non-linear effects of temperature on developmental rates), the development time is shorter at fluctuating temperatures than at constant temperatures, where the curve relating the development rate to temperature is accelerating (see Ragland and Kingsolver for details [45]). Given what is already known on the thermal ecology and thermal preferences of *D. plantaginea* and *A. matricariae*, the accelerating (convex) part of the curve is likely to be where the 7–17 °C (mean of 12 °C) regime stands. In addition, according to Kingsolver et al. [44], the shortened development time we observed in response to temperature fluctuations may depend on the amplitude of the tested variation, but may simply not be attributed to the Jensen’s inequality, as other complex time-dependent mechanisms can be at play.

The mass and size are strongly affected by developmental temperatures in ectotherms, and, according to our results, these traits seem to only be dependent upon the mean temperature experienced by the insect over the course of its ontogenesis [46]. Only for the mass of aphids was the fluctuating thermal regime different from the constant regime; aphids reared under the 7–17 °C fluctuating regime had a much higher mass than those reared under the constant 12 °C regime. These results are the opposite of what is found in *M. sexta* [44] or some fly species [47]. Again, this may be due to the Jensen’s inequality; the effect depends on the mean temperature and amplitude of the fluctuating regime, relative to the thermal limits of a species [29,47,48,49]. Our results underline that not all traits may respond the same way to fluctuating regimes (e.g., mass and size), because not all rate processes show identical performance curves (i.e., degree of thermal sensitivity of the process) and not all species may be affected by fluctuating regimes for a given trait.

Concerning longevity, there was a trend in both aphids and parasitoids towards increased survival rates under the fluctuating regime, compared to the constant 12 °C regime, but this trend was never confirmed statistically. As pointed out by Colinet et al. [29], fluctuating thermal regimes may decouple physiological age from chronological age, and species-dependent effects are likely to occur, resulting in the difficulty in predicting whether or not longevity would be affected by the type of temperature regime. To the same extent, we did not report any strong signal on the effect of fluctuating temperatures on aphid fecundity. Again, this issue is complex, as it may depend on the tested mean and thermal amplitude, and it may involve trade-offs with other traits that are also affected by temperature fluctuations [50,51]. In aphids, we obtained a much lower total fecundity at any temperature, as compared to Graf et al., who worked on the same species [23], probably because we used an artificial agar substrate to rear aphids on cut leaves.

The parasitism rate is intimately linked to parasitoid behavior and decision making, so it is not too surprising to see a strong beneficial effect of fluctuating regimes on this trait. Parasitoids may take advantage of the warmer periods of the day (17 °C) to parasitize aphids, and they may manage to compensate the inactivity (or decrease in activity) induced by low temperatures during the rest of the day. In the *Drosophila* larval parasitoid *Leptopilina boulardi* (Hymenoptera: Eucoilidae), the parasitism success is significantly higher under a fluctuating permissive temperature regime than at the constant corresponding mean temperature [43]. Again, these results are consistent with the predictions from Jensen’s inequality. Due to the fact that the temperature shows natural fluctuations over the day or over seasons, experiments based on constant temperatures could mask how thermal regimes really affect host–parasitoid interactions and dynamics [31].

Of course, in natural contexts, it is not only the temperature that varies over time, but also the humidity, light, food availability, plant quality, etc. These are parameters that are rarely studied in the laboratory under fluctuating conditions, because they are often considered less important than temperature, despite their great importance on the traits of individuals, on the dynamics of populations, and on species’ interactions within communities [52,53,54,55]. Finally, it is not only the fluctuation itself that is important to consider, but also the amplitude of the variation and the type of temperature change (smooth variations in a sinusoidal design, gentle slope in a triangle design like ours, or sharp change in a rectangle design), which are issues that should be further addressed in future studies [29,47]. Ultimately, both considering the set of environmental variations faced by insects and incorporating thermal fluctuation and extreme events in ecological and physiological studies would allow for the better informing of insect responses to climate change [56]. It would also improve pest outbreak models and the application of biological control strategies in the field; for example, by considering thermal shelters in orchards, or appropriate landscape management regarding microclimatic conditions.

## Figures and Tables

**Figure 1 insects-12-00855-f001:**
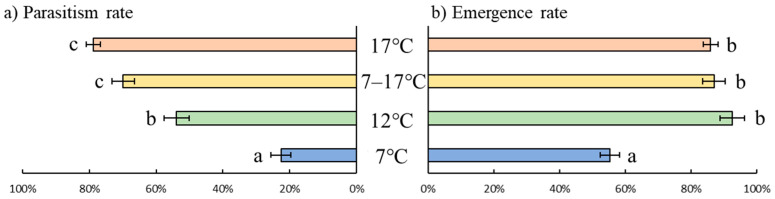
Parasitism rates (**a**) and emergence rates (**b**) of *Aphidius matricariae* parasitoids reared on *Dysaphis plantaginea* at three constant temperature regimes (7 °C, 12 °C, and 17 °C) and one fluctuating temperature regime (7–17 °C). Lowerscript letters indicate significant differences (*p* < 0.05) among temperature regimes for each measured parameter. N = 10.

**Figure 2 insects-12-00855-f002:**
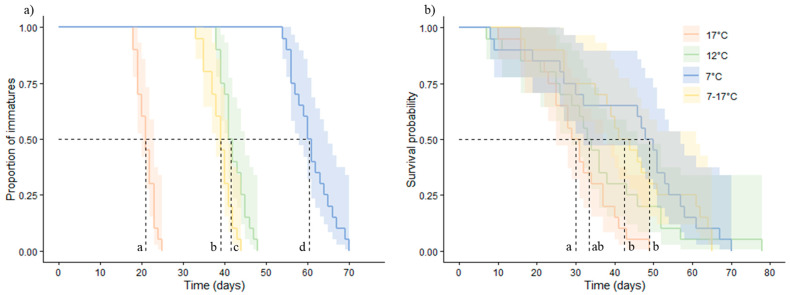
Proportion of immatures (i.e., development time to adulthood) (**a**) and adult survival probability (**b**) of *Aphidius matricariae* parasitoids exposed to three constant temperature regimes (7 °C, 12 °C, and 17 °C) and one fluctuating temperature regime (7–17 °C) for data of the two sexes pooled together. Dotted lines represent the median estimate for each temperature. Colored areas represent 95% CIs around estimates for each temperature. Lowerscript letters indicate significant differences (*p* < 0.05) among temperature regimes for each measured parameter. N = 20.

**Table 1 insects-12-00855-t001:** Life parameters (mean ± SE) for *Dysaphis plantaginea* aphids exposed to three constant temperature regimes (7 °C, 12 °C, and 17 °C) and one fluctuating temperature regime (7–17 °C). N = 12 for all parameters, except for the pre-reproductive period, because some aphids did not reach the adult stage (N indicated in brackets for this parameter). Lowerscript letters indicate significant differences (*p* < 0.05) among temperature regimes for each measured parameter.

Temperature Regime	Pre-reproductive Period (Days)	Longevity (Days)	Total Offspring	Immature Mortality Level (%)
Constant 7 °C	37.5 ± 1.3 [8] ^d^	51.0 ± 8.3 ^b^	22.0 ± 4.1 ^a^	33.3
Constant 12 °C	27.9 ± 0.8 [9] ^c^	36.1 ± 4.2 ^ab^	31.4 ± 4.7 ^b^	25.0
Constant 17 °C	15.4 ± 0.7 [9] ^a^	29.3 ± 3.3 ^a^	44.7 ± 6.3 ^c^	25.0
Fluctuating 7–17 °C	21.0 ± 0.6 [10] ^b^	40.7 ± 1.9 ^ab^	34.1 ± 4.2 ^bc^	16.7

**Table 2 insects-12-00855-t002:** Mass and tibia size (mean ± SE) of *Aphidius matricariae* parasitoids, detailed for both males (♂) and females (♀), and of *Dysaphis plantaginea* aphids, exposed to three constant temperature regimes (7 °C, 12 °C, and 17 °C) and one fluctuating temperature regime (7–17 °C). N = 20, except for aphids at 7 °C, where N = 13. Lowerscript letters indicate significant differences (*p* < 0.05) among temperature regimes for each measured parameter. Stars (*) indicate significant differences (*p* < 0.05) between males and females for a given temperature regime.

Parasitoids			Aphids	
Temperature Regime	Mass (µg)	Size (mm)	Mass (µg)	Size (mm)
Constant 7 °C	116.0 ± 8.4 ^b^	0.59 ± 0.03 ^b^	111.8 ± 7.7 ^b^	1.20 ± 0.04 ^b^
	(♂113.6 ± 12.6; ♀118.3 ± 11.6)	(♂0.56 ± 0.04; ♀0.63 ± 0.03)		
Constant 12 °C	103.5 ± 7.2 ^ab^	0.52 ± 0.03 ^ab^	82.7 ± 4.7 ^a^	1.12 ± 0.02 ^b^
	(♂95.1 ± 10.8; ♀113.8 ± 8.2)	(♂0.48 ± 0.04; ♀0.57 ± 0.02) *		
Constant 17 °C	91.2 ± 5.5 ^a^	0.46 ± 0.02 ^a^	80.6 ± 3.9 ^a^	0.94 ± 0.05 ^a^
	(♂80.8 ± 9.8; ♀98.1 ± 5.9)	(♂0.41 ± 0.03; ♀0.50 ± 0.03) *		
Fluctuating 7–17 °C	103.1 ± 6.4 ^ab^	0.50 ± 0.02 ^ab^	101.5 ± 3.1 ^b^	1.08 ± 0.01 ^b^
	(♂97.9 ± 11.2; ♀107.8 ± 7.4)	(♂0.47 ± 0.04; ♀0.52 ± 0.03)		

## Data Availability

Data have been made publicly available and can be retrieved at the following doi:10.5281/zenodo.5342664.

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
