# Peer review of "Effects of Constant versus Fluctuating Temperatures on Fitness Indicators of the Aphid *Dysaphis plantaginea* and the Parasitoid *Aphidius matricariae"

_insects, 2021, doi:10.3390/insects12100855_

Round 1

Reviewer 1 Report

The ms is well-written and the working hypothesis clearly presented. The quantity and quality of data presented are adequate to support authors' hypothesis. The statistical analysis used seems appropriate for this kind of data and the results clearly presented and adequately discussed according to the published literature.

I have made a few comments as explained below.  

L153: I think this (“the three treatments”) is wrong.

Tables and figures: Lower script letters for significant difference are disorder.

Throughout the ms, the authors should be used the same order (from lower to higher or vice versa).

Author Response

Rev 1:

The ms is well-written and the working hypothesis clearly presented. The quantity and quality of data presented are adequate to support authors' hypothesis. The statistical analysis used seems appropriate for this kind of data and the results clearly presented and adequately discussed according to the published literature.

We would like to thank Reviewer 1 for the very positive input on our manuscript, and for the suggested modifications.

I have made a few comments as explained below.  

L153: I think this (“the three treatments”) is wrong.

Replaced by “The four treatments”, thanks for noticing this mistake.

Tables and figures: Lower script letters for significant difference are disorder.

Throughout the ms, the authors should be used the same order (from lower to higher or vice versa).

Lower script letters have been rearranged according to their value. The letter a comes from the lowest value.

Reviewer 2 Report

An interesting manuscript, of scientific and practical interest. Please see my suggestions attached to this message. 

Reviewer 3 Report

The aims are clear and clearly stated, the design is appropriate to reach the aims. I have only a few minor formal recommendations for a better presentation of the results. I also made a few remarks to the discussion, although many of the issues seem to be somewhat treated later in the discussion.

Line 89: provide LDT and other quantitative parameters.

Line 93: "longer growth rates" - rate can be lower, time can be longer.

Lin 111: 15°C - provide the range.

Line 148: Decsribe that it was not a rectangular alternation of two temperatures, as usual, but a triangle-like gradual decrease and increase regime.

Line 238 and elsewhere: In the table and figure captions I recommend writing the genus in full (for quick readers that do not read all text).

Table: Prereproductive period: better to arrange the letters ordered according to the value.

Line 274 and elsewhere: "confounded" -> merged or pooled or joint

Line 278: You can calculate LDT and SEM and compare the two parameters with literature (line 318). 

Figure 2: I strongly recommend keeping the same colour scheme for all figures. Better is the scheme in Fig. 1.

Line 292: insert "Lowerscript letters "

Line 293: "N=10" - No. N varies, original N was higher. 

Line 311: Slightly correct the sentence. The bigger size is not necessarily CAUSED by longer developmental time (growth rate cannot be longer, it is lower or higher, and growth is not development). The arthropod body has some measurement mechanisms to stop the growth and moult, and the growth rate and developmental rate are different phenomena. 

Line 318: "indicating" - Correct /delete. The lower thresholds for various processes need not correlate too much with the developmental threshold.

Line 328: Two of how many? Better to write the percentage. 

Line 358: You may mention the possible effect of your triangle fluctuating regime and rectangle alternating of two temperatures. 

Line 380: Again. Mention triangle vs. rectangle or sinusoid shapes of variation.

Author Response

Rev 3: Please find our responses in bold.

The aims are clear and clearly stated, the design is appropriate to reach the aims. I have only a few minor formal recommendations for a better presentation of the results. I also made a few remarks to the discussion, although many of the issues seem to be somewhat treated later in the discussion.

We would like to thank Reviewer 3 for the very positive comments on our manuscript, and for the suggested modifications.

Line 89: provide LDT and other quantitative parameters.

We have added: “Lower development threshold is 4.5°C in D. plantaginea and around 5°C in A. matricariae [20, 24].”
Other quantitative parameters have been mentioned in the same paragraph, but we did not want to add too much details in the introduction of our manuscript, because precise details can be found in the cited articles.

Line 93: "longer growth rates" - rate can be lower, time can be longer.

Thanks for pointing that out. Indeed, low temperature leads to reduced growth rate so longer development time. We have changed the sentence to: “because of reduced growth rates leading to longer development times”.

Lin 111: 15°C - provide the range.

Done. We have added (±10°C)

Line 148: Decsribe that it was not a rectangular alternation of two temperatures, as usual, but a triangle-like gradual decrease and increase regime.

The sentences have been changed to: “fluctuating regime of 7-17°C (12h), 17-7°C (12h), under a 12:12 h photoregime, corresponding to a mean temperature of 12°C over the day and a rate of temperature change of 1.66°C per hour. For the fluctuating regime, the 12h increasing temperature phase started at mid-night and the 12h decreasing temperature phase started at mid-day, resulting in a triangle-like gradual decrease and increase regime.”

Line 238 and elsewhere: In the table and figure captions I recommend writing the genus in full (for quick readers that do not read all text).

Done

Table: Prereproductive period: better to arrange the letters ordered according to the value.

Lower script letters have been rearranged according to their value in Tables and Figures. The letter a comes from the lowest value.

Line 274 and elsewhere: "confounded" -> merged or pooled or joint

Done

Line 278: You can calculate LDT and SEM and compare the two parameters with literature (line 318). 

LDT cannot be accurately calculated from our dataset, because we have too few datapoints (i.e., too few thermal regimes). We have run another (ongoing) experiment to better assess LDT in D. plantaginea and A. matricariae under various thermal conditions.

Figure 2: I strongly recommend keeping the same colour scheme for all figures. Better is the scheme in Fig. 1.

Done. We have modified the colors from figure 2a and 2b. They now match colors from Fig1.

Line 292: insert "Lowerscript letters "

Done

Line 293: "N=10" - No. N varies, original N was higher. 

We have changed the sample size to N = 20, because as stated in the material and methods section, we took 2 parasitoids per brood (i.e. per maternal origin), and there were 10 broods. So N = 20 parasitoids tested for longevity per treatment and represented on this figure.

Line 311: Slightly correct the sentence. The bigger size is not necessarily CAUSED by longer developmental time (growth rate cannot be longer, it is lower or higher, and growth is not development). The arthropod body has some measurement mechanisms to stop the growth and moult, and the growth rate and developmental rate are different phenomena. 

We have replaced the sentence by: “which could be in part because of higher growth rates”

The reviewer is right on this point, and on the differences between development and growth, and we are aware that we have made a semantic shortcut. In this sentence, what is important is the generally observed result that arthropods are larger at lower temperatures (although the reverse is also possible). The underlying mechanisms cannot really be discussed in this article because we have not specifically studied them.

Line 318: "indicating" - Correct /delete. The lower thresholds for various processes need not correlate too much with the developmental threshold.

This is a good point. The end of the sentence has been modified to: “suggesting that some physiological or behavioural processes could have been impaired if their lower threshold is as close to these temperature values”

Line 328: Two of how many? Better to write the percentage. 

We have added “25 to 23 days”. We have also added the amplitude of the fluctuating regime used in the cited study: +/- 15°C.

Line 358: You may mention the possible effect of your triangle fluctuating regime and rectangle alternating of two temperatures. Line 380: Again. Mention triangle vs. rectangle or sinusoid shapes of variation.

We have added: “and the type of temperature change (smooth variations in a sinusoidal design, gentle slope in a triangle design like ours, or sharp change in a rectangle design),” L395